# Investigation of preference for local and global processing of Capuchin-monkeys (*Sapajus* spp.) in shape discrimination of mosaic arrangements

**Fernanda Mendes**[1,2]*, **Ana Leda de Faria Brino**[1], **Paulo Roney Kilpp Goulart**[1], **Olavo de Faria Galvão**[1], **Dora Selma Fix Ventura**[3], **Letícia Miquilini**[1], **Felipe André da Costa Brito**[4], **Givago Silva Souza**[2,4]

**1** Núcleo de Teoria e Pesquisa do Comportamento, Universidade Federal do Pará, Belém, Brazil, **2** Núcleo de Medicina Tropical, Universidade Federal do Pará, Belém, Brazil, **3** Instituto de Psicologia, Universidade de São Paulo, São Paulo, Brazil, **4** Instituto de Ciências Biológicas, Universidade Federal do Pará, Belém, Brazil

* fmendesr13@gmail.com

**Data Availability Statement:** All relevant data are within the manuscript and its Supporting Information files.

## Abstract

Classical experiments using hierarchical stimuli to investigate the ability of capuchin monkeys to integrate visual information based on global or local clues reported findings suggesting a behavioral preference for local information of the image. Many experiments using mosaics have been conducted with capuchin monkeys to identify some of their perceptual phenotypes. As the identification of an image in a mosaic demands the integration of elements that share some visual features, we evaluated the discrimination of shapes presented in solid and mosaic stimuli in capuchin monkeys. Shape discrimination performance was tested in 2 male adult capuchin monkeys in an experimental chamber with a touchscreen video monitor, in three experiments: (i) evaluation of global and local processing using hierarchical stimuli; (ii) evaluation of target detection using simple discrimination procedures; (iii) evaluation of shape discrimination using simple discrimination and delayed matching-to-sample procedures. We observed that both monkeys had preferences for local processing when tested by hierarchical stimuli. Additionally, detection performance for solid and mosaic targets was highly significant, but for shape discrimination tasks we found significant performance when using solid figures, non-significant performance when using circle and square shapes in mosaic stimuli, and significant performance when using Letter X and Number 8 shapes in mosaic stimuli. Our results are suggestive that the monkeys respond to local contrast and partly to global contrast in mosaic stimuli.

## Introduction

Research on perceptual grouping has extensively examined two levels of hierarchical processing, centering on the relative primacy of global features versus local features within a scene [1–

**Funding:** DFV, GSS, OFG #431748/2016-0 Conselho Nacional de Desenvolvimento Científico e Tecnológico (CNPq) https://www.gov.br/cnpq/pt-br Funding through FM, FACB, LM Finance Code 001 Coordenação de Aperfeiçoamento de Pessoal de Nível Superior (CAPES) https://www.gov.br/capes/pt-br.

**Competing interests:** The authors have declared that no competing interests exist.

6]. Global processing involves integrating various elements into a coherent whole, while local processing concentrates on the finer details of a prominent object or situation [7, 8].

In his research, Navon[8] examined the role of global and local cues in the perceptual organization of images in humans, utilizing hierarchical stimuli where larger global shapes, often letters, were formed by smaller local elements. These global and local shapes could either be identical (consistent stimulus) or differ (inconsistent stimulus). His findings demonstrated that human perception exhibited a global advantage or precedence, meaning that individuals tended to perceive the overall global structure before discerning the specific local elements constituting the scene. Examples of hierarchical stimuli are shown in the Fig 1.

Numerous studies exploring perceptual grouping processes with visual stimuli reveal variations in the precedence of global or local cues across different species, including humans [9–11], chimpanzees [3, 11, 12], baboons [10], rhesus monkeys [12], cotton top tamarins [13], and capuchin monkeys [6, 9, 14]. In the context of platyrrhine monkeys, investigations into global and local processing have predominantly focused on capuchin monkeys [1, 5, 6, 9, 14–19]. Several of these experiments utilized matching-to-sample (MTS) procedures, and their outcomes consistently demonstrated an advantage for local level of hierarchical processing [1, 14, 16, 18, 19].

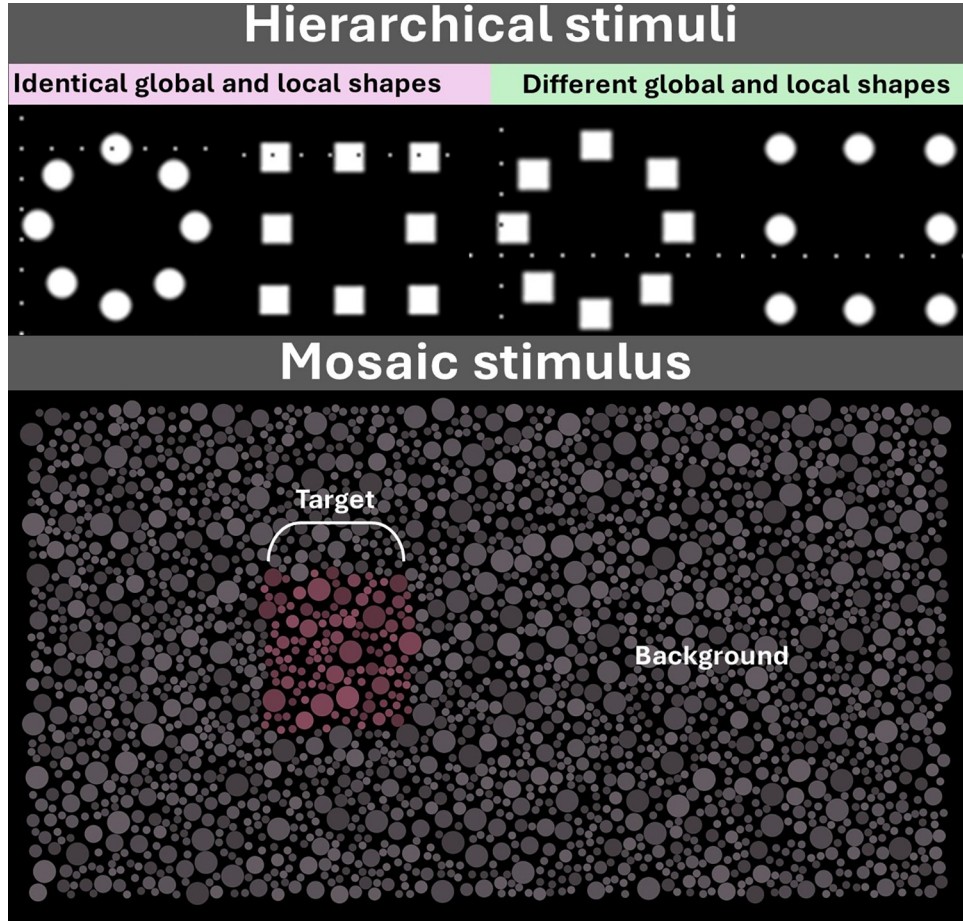

**Fig 1. Hierarchical and mosaic stimulus.** Hierarchical stimuli have identical and different global and local clues that can be used for visual processing. Mosaic stimuli is composed of non-overlapping elements and a subgroup of the mosaic elements shares some feature (in the case of the figure the shared feature is the chromaticity) that creates an illusory target over a background.

In recent years, researchers have conducted studies to investigate the chromatic discrimination abilities of platyrrhine monkeys using a stimulus named pseudoisochromatic [20, 21], which is formed by a set of non-overlapped individual elements arranged as a mosaic of circular patches. A subgroup of these elements shared the same color shaping a visual target over the remained elements that had different color compared to the target in region of the stimulus named background. Although there is no overlapping in the different elements from target of the mosaic, there is a construction of illusory object because the similarity of colors they share (Fig 1). For instance, studies have been conducted with squirrel monkeys (*Saimiri* spp.: [22]), capuchin monkeys (*Sapajus* spp.: [20,21]), and howler monkeys (*Alouatta* spp.: [23]) using mosaics which the target is an illusory square. In these experiments, the subjects' successful performance necessitates identifying the chromatic target irrespective of its shape. It remains unclear whether non-human primates select the target by its identification as global object or by detecting a region of the mosaic with different color from the other parts of the mosaic.

The present study aimed to extend the question of preference for global versus local processing in capuchin monkeys by using mosaic stimuli. We assumed that the perception of the target in mosaic stimuli would represent the end product of global processing and that shape discrimination would be an appropriate method to investigate the emergence of the illusory target from mosaics. The absence of shape discrimination favored the hypothesis of a preference for local processing in these monkeys. For that, we examined both local and global processing through hierarchical stimuli as a control experiment, as well as target detection involving solid and mosaic stimuli.

## Methods

### Subjects

Two adult male capuchin monkeys (*Sapajus* spp.), namely Tico and Raul, were involved in the study (Fig 2A and 2B). Both monkeys possessed significant prior experience with simple discrimination and matching-to-sample procedures, like those utilized in the present research [20, 24–28]. Moreover, Tico had previously participated in experiments focused on tool use [29]. For the record, the subjects were genetically and behaviorally identified as dichromat deuteranopes, as established by previous research [20].

They were housed in nursery cages along with one or two other Sapajus in an open area at the Experimental School for Primates, which is located at the Federal University of Pará. Tico shared his living space with another male monkey in a glass-enclosed housing measuring 45.09 m$^2$. This glass enclosure lacked visual barriers and a roof, and it was situated next to an

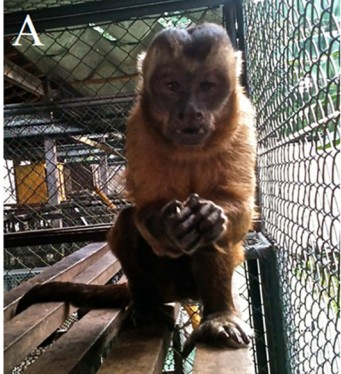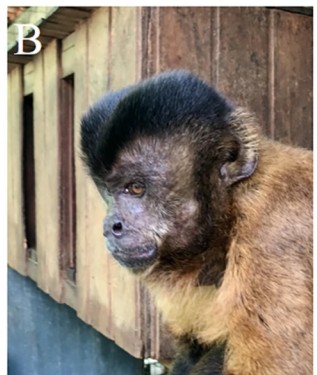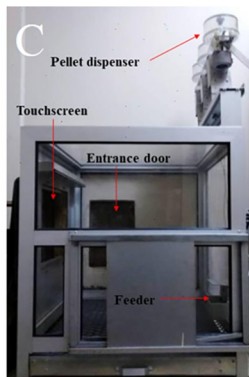

**Fig 2.** Subjects Tico (A) and Raul (B), and experimental chamber (C).

identical enclosure that housed two other monkeys. The two enclosures were separated by a glass wall and had partial coverage to protect the primates from direct sunlight. Raul resided with two other monkeys in a metal nursery that measured 3.00 m x 2.50 m x 2.50 m (height x width x length). The nursery rested on a cement floor and provided partial coverage against direct sunlight. All enclosures were equipped with drinking water fountains featuring automatic spouts. The primates' diet consisted of a special ration for primates from Quimtia (Nuvilab neotropical primates, extruded food), which was distributed twice a day, once in the early morning and once in the late afternoon. Additionally, the animals received a meal at 3:00 pm daily, consisting of fruits, eggs (three times a week), and vegetables. Before the experiments the subject were kept at a certain percentage of free-feeding weight. The Experimental School for Primates holds an operating license (LO N° 12493/2021) that categorizes it as a scientific breeding ground for research purposes. This license was issued by the local environmental agency (SEMAS—Secretaria Estadual de Meio Ambiente e Sustentabilidade [State Secretariat for the Environment and Sustainability]), which is responsible for inspecting environmental management activities. Furthermore, the experimental procedures conducted on the primates received approval from the Ethics Committee on Animal Use of the Federal University of Pará (CEUA/UFPA), with protocol number #4629251121.

## Apparatus

For the purpose of data collection, we utilized an experimental chamber measuring 0.60 m x 0.60 m x 0.60 m. This chamber was equipped with a 15-inch color LCD monitor (Elo Systems®, 18 inches) with a surface acoustic wave touchscreen-type device that could be accessed through a rectangular opening (0.26 m long x 0.20 m high) in the front wall of the chamber (refer to Fig 2C). The control of the experimental sessions, the pellet dispenser, and the recording of the monkey's responses were managed by a home-made software programmed in C++. The pellet dispenser was triggered upon the occurrence of correct choices, releasing pellets into a tray positioned on the opposite side of the screen. Positioned just above the corners of the touchscreen, two small red lamps would illuminate upon a correct response. Additionally, it facilitated the generation of scheduled sessions, recorded response data, and generated a comprehensive report encompassing details such as the date, time, subject, and task performance.

## Experimental procedures

To explore the impact of perceptual global and local processing on the shape discrimination capabilities of capuchin monkeys using a mosaic design, we carried out a series of three experiments: (i) evaluating the capuchins' proficiency in processing the global and local attributes of hierarchical visual stimuli as outlined in Spinozzi et al. [18] (ii) appraising target detection through both solid and mosaic stimuli; and (iii) discerning shape discrimination utilizing stimuli presented in mosaic design as well as solid stimuli. Each subject performed one session (48 trials) per day.

We wrote a routine in Python language to calculate the statistical significance of the subject's performance in each experiment by calculating the binomial probability, which was the probability to reach $x$ or more successful decisions in $n$ repeated trials in an experiment with z possible decisions and success probability in a single trial of p equal to 1/z (Eqs 1 and 2). We considered as significant number of hits in the experiment when the binomial probability was smaller than 0.05.

$$bp = {_nC_x} \times p^x \times (1-p)^{n-x} \tag{Eq1}$$

$$_nC_x = \frac{(n!)}{x! \times (n-x)!} \tag{Eq2}$$

To facilitate the comprehension, we chose to describe the methods of each experiment followed by its results.

### Experiment 1. Global and local processing evaluation

**Stimuli.** In the context of a delayed matching-to-sample task, we employed four hierarchical stimuli reminiscent of those utilized by Spinozzi et al. [18]. These stimuli comprised configurations spanning 4.9˚ of visual angle, each encompassing 8 individual circles or squares (each with a diameter or side measuring 0.95˚), as illustrated in Fig 3. Depending on the interplay between shapes at the global and local levels, a stimulus was designated as either "Consistent" (characterized by congruent global and local shapes) or "Inconsistent" (marked by incongruent global and local shapes).

**Procedure.** Four distinct configurations of delayed matching-to-sample training were executed, encompassing: (1) the global condition with consistent trials, (2) the global condition with inconsistent trials, (3) the local condition with consistent trials, and (4) the local condition with inconsistent trials, all of which are depicted in Fig 3. The animals were exposed to five sessions with global configuration trials and another five sessions with local configuration trials. The trials were repeated and there were differential consequences for success and error. Five sessions with 48 trials had global configurations, in a total of 240 trials, 120 of global consistent and 120 of global inconsistent; the same occurred for the five sessions of 48 trials having local configuration, 240 total trials, 120 local consistent and 120 local inconsistent.

The two-choice 1-second delayed matching-to-sample procedure was applied. There were only two possible stimuli. At the initiation of each trial, a sample stimulus appeared on a dark screen, prompting the monkey to initiate the task by tapping twice on the sample. A correct response from the monkey prompted a 1-second delay, followed by the presentation of one S+ (equal to the sample) and one S- stimuli (different to the sample) on a dark screen. The trial concluded upon the monkey's one tap on any stimulus (S+ or S-). If the monkey's tap fell upon the S+, a banana pellet reward was dispensed. The intertrial interval persisted for 6 seconds, during which the monitor screen remained dark. Touches on the dark screen during this interval held no consequences.

The trial background and other features were identical in the global and local configurations. The experimenter defined the rule related to each configuration trial and rewarded the animal whenever it followed that rule. In this study, the global trials have different comparisons in terms of the global shape, having the same local shapes, so that to choose the correct comparison, the subject had to look at the global aspect. In trials called local, the comparisons had identical global shapes and distinct local shapes so that the subject could only choose the correct one if they were responding based on local shape. The order of stimulus presentation was randomly chosen by the software in each session. The monkeys were instructed on which global and local clues to follow using a reward system that favored each condition.

The subjects were exposed to the two conditions sequentially. Firstly, Raul underwent five sessions of the global condition, while simultaneously Tico engaged in five sessions of the local condition. Subsequently, Tico underwent five training sessions in the global condition, while Raul participated in the five training sessions of the local condition. The evaluation of subjects' performance was based on the accuracy of responses across the five sessions.

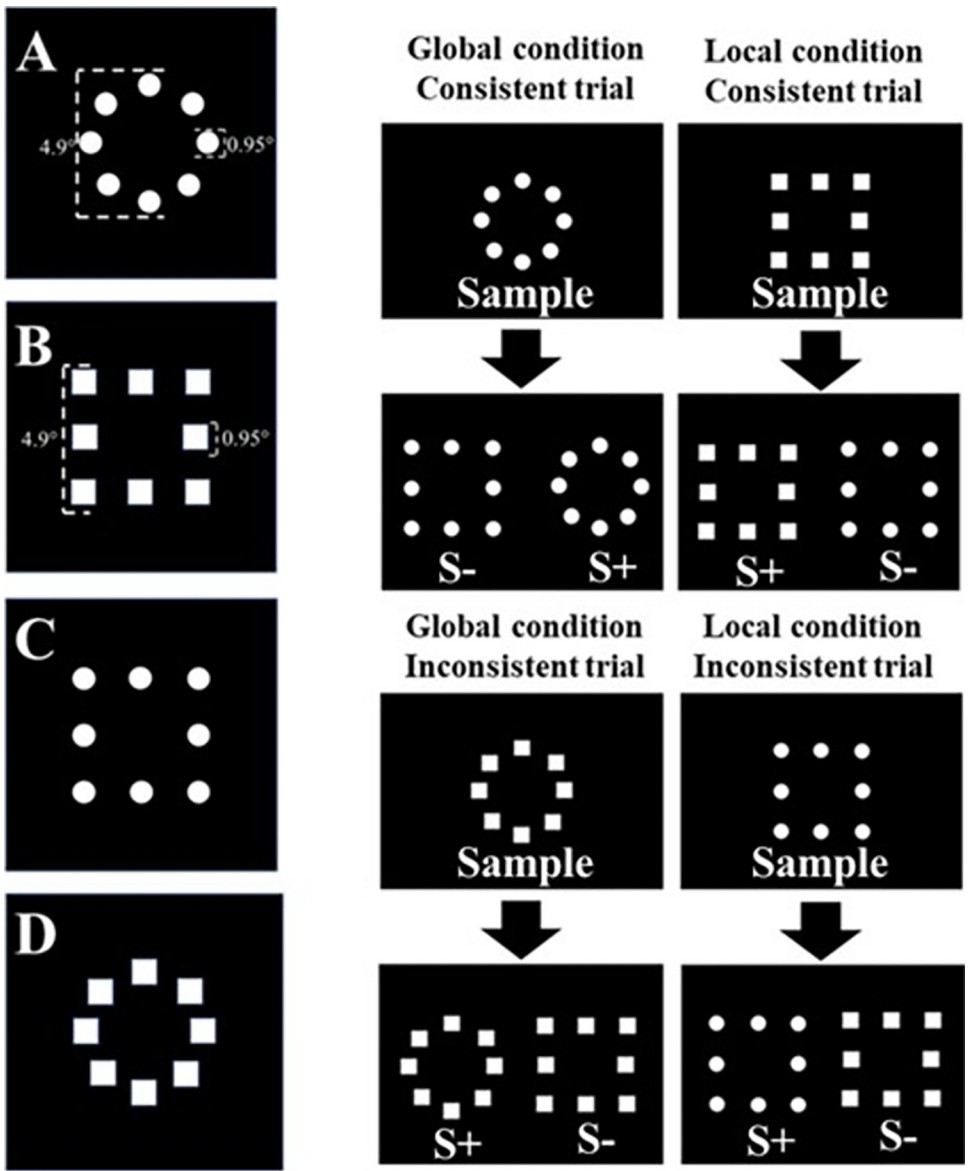

**Fig 3. Global and local conditions with hierarchical stimuli.** Hierarchical stimuli were made up of eight local elements. Local elements (circles or squares) could be consistent as in A and B or inconsistent as in C and D with the global shape of the stimulus. The experiment comprised a delayed matching-to-sample test using four different configurations: global condition and consistent trial, global condition and inconsistent trial, local condition and consistent trial, and local condition and inconsistent trial are exemplified in the figure. Sample: model to search for in the next window; S+: correct choice; S-: incorrect choice.

## Results of the Experiment 1

### Results of the global and local processing evaluation

Table 1 presents the performance data, represented as the percentage of hits (correct responses out of total trials), exhibited by both subjects during the trials involving global and local processing. Notably, we observed that both subjects demonstrated nonsignificant performance levels in the experiments pertaining to global processing, encompassing both consistent and inconsistent trials. Conversely, in the experiments related to the local condition, encompassing

**Table 1. Performance for global and local processing from both subjects.** The statistical significance was based on probability of success = 0.5, number of trials, and number of successes. The values represent the number of successes per number of trials.

| Test condition | Raul | p-value | Tico | p-value |
|---|---|---|---|---|
| Global condition and consistent trial | 69/120 | 0.06 | 66/120 | 0.15 |
| Global condition and inconsistent trial | 65/120 | 0.2 | 50/120 | 0.97 |
| Local condition and consistent trial | 104/120 | <0.001* | 107/120 | <0.001* |
| Local condition and inconsistent trial | 101/120 | <0.001* | 102/120 | <0.001* |

*Significant number of successes.

both consistent and inconsistent trials, the subjects displayed statistically significant performance. These results suggest a preference for local processing in capuchin monkeys.

## Experiments 2 and 3: Assessment of stimulus target detection and shape discrimination

Two sets of experiments were conducted to elucidate the subjects' capacity for target detection (Experiment 2) and shape discrimination utilizing both mosaic and solid stimuli (Experiment 3). To achieve this, we employed an identical set of stimuli as delineated below.

## Stimuli

Mosaic stimuli were meticulously crafted within the MATLAB programming environment (R2021a, Mathworks, CA). Each stimulus comprised 1100 circles across four distinct sizes: 0.54, 0.44, 0.32, and 0.22 degrees of visual angle. The spacing between circles was deliberately randomized with no overlap among them. Conversely, solid stimuli were meticulously fashioned in commercial image editor (Microsoft Paint, Microsoft, Palo Alto, USA). In each set of stimuli (mosaic and solid sets), a diverse array of six targets—circle, square, X letter, star, musical keynote, and number eight—was incorporated, along with a background figure. For both solid and mosaic stimuli, uniform chromaticity (CIE 1976 color space: u' = 0.219; v' = 0.48) was upheld, ensuring high luminance contrast exceeding 90% Weber contrast.

## Experiment 2: Target detection

We employed a simple discrimination procedure to assess the capability of capuchin monkeys in detecting a target (S+) among eight other stimuli (S-) arranged in a 3 x 3 grid. This procedure was conducted employing both solid and mosaic stimuli, as depicted in Fig 4. We performed this evaluation in three stages. The first stage was to evaluate the detection of six distinct solid target in a session, encompassing 8 trials for each S+ per session. This stage was completed when the subject reached minimum 90% correct in one session. The order of targets presentation was randomly chosen in each session.

Subsequently, in the second stage, six simple discrimination sessions were carried out, and in each session a solid target stimulus was replaced by its correspondent mosaic target stimulus while retaining the other solid stimuli. In each session, there were 8 trials per target. The order of stimulus replacement was randomly chosen by the software in each session. Finally, the third stage involved a final simple discrimination session entailed 8 trials for each target in the mosaic.

For all stages, a correct response, acknowledged with a banana pellet reward, was defined as a tap on the S+. Conversely, a tap on any S- was considered an erroneous response, resulting in no reward release. After the subject response, a dark screen was displayed for 6 seconds

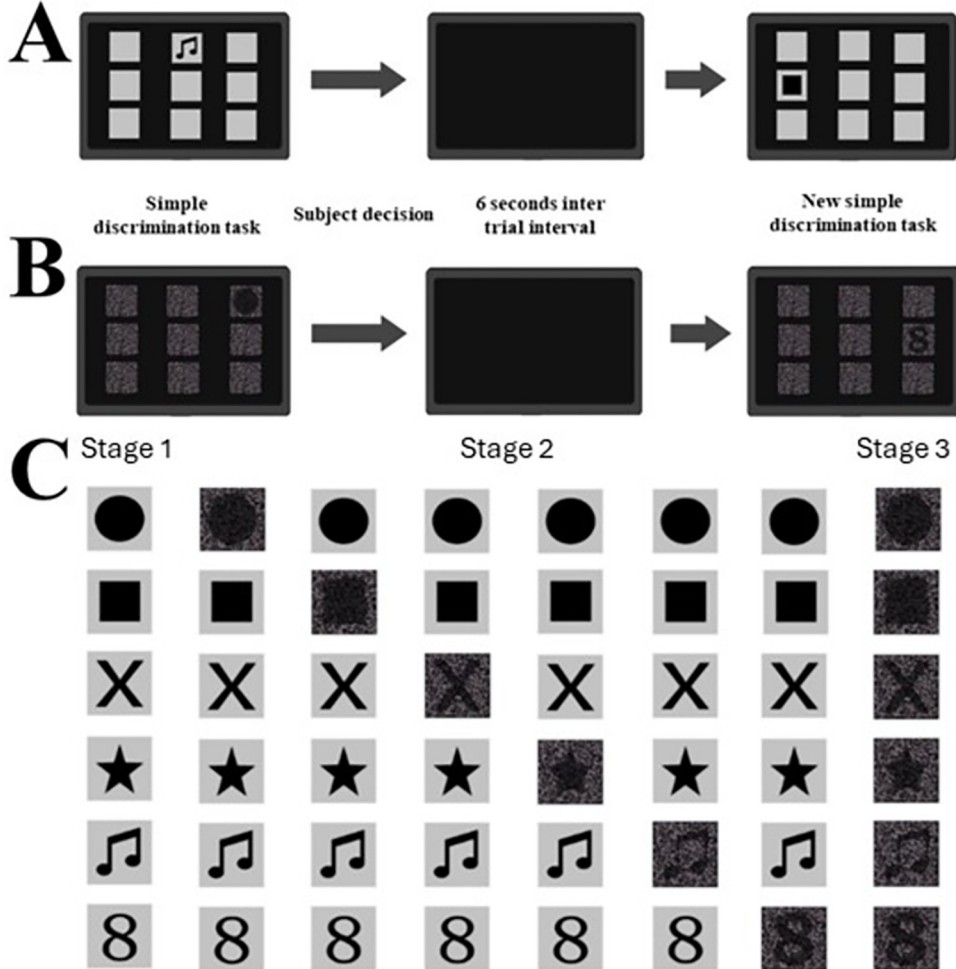

**Fig 4. Simple discrimination task for stimulus target detection.** The same procedure was carried out using solid (A) and mosaic (B) stimuli. After the stimulus presentation a touch on the S+ was considered a correct response, and a pellet was delivered to the animal, a touch on the S- was considered an incorrect response and no reward was released. The touch on S+ or S- leads to a dark screen of 6 s (intertrial interval) followed by a new stimulus for simple discrimination trial. (C) Rationale used to guide the trials.

(intertrial interval), followed by the presentation of a new stimulus for a fresh simple discrimination trial. Notably, touching the dark screen background during the trial presentation or intertrial interval bore no consequences. The predetermined criteria for completion of the test encompassed achieving either a 90% accuracy rate in one session or concluding 10 sessions without reaching the criterion. The same order of sessions was delivered for both animals, and inside each session the order of stimulus presentation was randomly chosen by the software.

## Results of the Experiment 2

### Assessment of stimulus target detection

Considering all the stages, Tico carried out 48 trials per solid target stimulus and 16 trials per mosaic target stimulus, and Raul carried out 72 trials per solid target stimulus and 16 per mosaic target stimulus. The difference between the number of trials per solid target stimulus carried out by the subjects occurred because Tico and Raul completed the first stage in 1 and 3

**Table 2. Performance for target detection from both subjects.** The statistical significance was based on probability of success = 0.11, number of trials, and number of successes. The values represent the number of successes per number of trials.

| Stimulus | Raul | | | | Tico | | | |
|---|---|---|---|---|---|---|---|---|
| | Solid | p-value | Mosaic | p-value | Solid | p-value | Mosaic | p-value |
| Circle | 69/72 | <0.001* | 16/16 | <0.001* | 45/48 | <0.001* | 15/16 | <0.001* |
| Square | 67/72 | <0.001* | 15/16 | <0.001* | 48/48 | <0.001* | 15/16 | <0.001* |
| Letter X | 68/72 | <0.001* | 15/16 | <0.001* | 46/48 | <0.001* | 16/16 | <0.001* |
| Star | 65/72 | <0.001* | 12/16 | <0.001* | 45/48 | <0.001* | 16/16 | <0.001* |
| Musical keynote | 68/72 | <0.001* | 15/16 | <0.001* | 47/48 | <0.001* | 14/16 | <0.001* |
| Number 8 | 60/64 | <0.001* | 20/24 | <0.001* | 45/48 | <0.001* | 16/16 | <0.001* |

*Significant number of successes.

sessions, respectively. Table 2 illustrates the performance outcomes of Raul and Tico in the context of target detection within a simple discrimination task across all the stages. Notably, both subjects exhibited significant performance levels across both solid and mosaic stimuli, thereby indicating their adeptness at detecting the target within both design formats.

## Experiment 3. Shape simple discrimination

The third experiment implemented a simple discrimination procedure to investigate the discernment capabilities of capuchin monkeys concerning a stimulus featuring a distinct target (S +) set apart from eight identical targets (S-) within a 3x3 array, as depicted in Fig 5. This procedure was executed employing either solid or mosaic stimuli. The methodology for assessing shape discrimination retained the same underlying rationale as applied in the target detection experiment. Initially, a shape discrimination session employing solely solid stimuli was conducted, encompassing 48 trials with 8 trials per S+. A sequence of shape simple discrimination sessions ensued until the subject had a minimum performance of 90% correct in one session. After criterion, one array featured a mosaic design were presented, one by one, with remaining five arrays retained a solid design until all six mosaics were presented. In case the subject performed 10 consecutive sessions without reach the criterion, the test would be completed. The order of targets presentation was randomly chosen by the software.

Consistent with the preceding experiment, a tap on the S+ signified a precise response and merited a banana pellet reward, whereas a tap on the S- signified an erroneous response, resulting in no reward dispensed. In case of tap occurred on either the S+ or the S-, a dark screen emerged for 6 seconds (intertrial interval), followed by the commencement of a new shape discrimination task. Taps on the dark screen during trial presentation or the intertrial interval held no impact.

Another experiment involving the matching-to-sample paradigm was undertaken to assess shape discrimination capabilities. This procedure aimed to ascertain whether capuchin monkeys could successfully match identical shapes presented through solid or mosaic stimuli. Each session encompassed 54 trials, consisting of 9 trials for each S+. At the initiation of each trial, a sample stimulus appeared on a dark screen, prompting the monkey to initiate the task by tapping twice on the sample. A correct response from the monkey prompted a 1-second delay, followed by the presentation of one S+ and two S- stimuli on a dark screen. The trial concluded upon the monkey's tap on any stimulus (S+ or S-). If the monkey's tap upon the S+, a banana pellet reward was dispensed. The intertrial interval persisted for 6 seconds, during which the monitor screen remained dark. Again, taps on the dark screen during this interval held no consequences. The procedural details of this experiment are illustrated in Fig 5. The performance criterion for terminating a session was either achieving a 90% accuracy rate or the

## Shape discrimination test using simple discrimation procedure

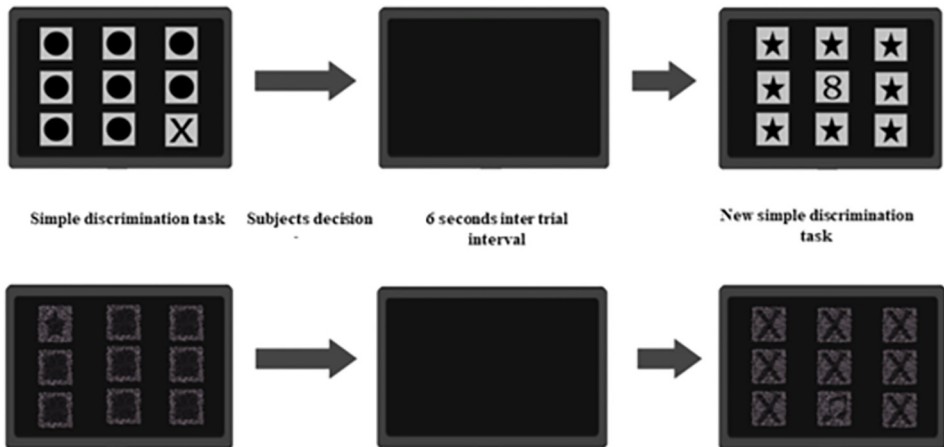

## Shape discrimination test using delayed matching to sample procedure

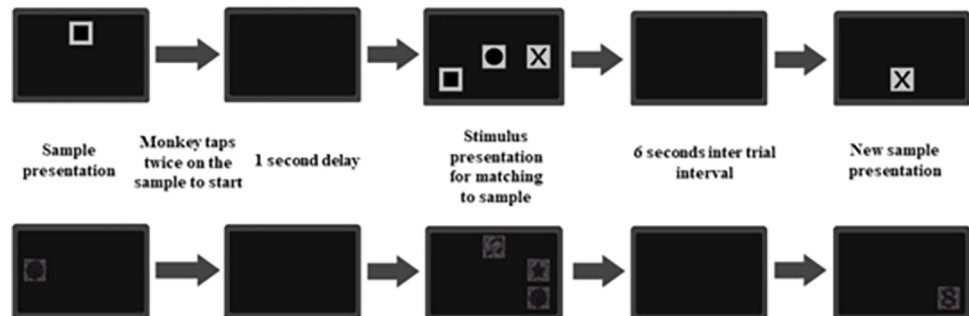

**Fig 5.** Simple discrimination (upper lines) and matching-to-sample (lower lines) procedures used to assess shape discrimination. The same procedures were carried out using solid and mosaic stimuli.

completion of 5 sessions. For all conditions, the order of stimulus presentation was randomly chosen by the software in each session.

## Results of the Experiment 3

### Assessment of shape discrimination: Simple discrimination procedure

Presented in Table 3 are the performance outcomes of the subjects in the context of the shape discrimination test, conducted through a simple discrimination procedure. Raul reached performance criterion (90% correct) before Tico, so the total number of trials with solid stimuli was lower for Raul (96 trials distributed in 12 sessions) than for Tico (120 trials distributed in 15 sessions). Strikingly, both subjects exhibited superior performance when faced with solid figures as stimuli, achieving significance across all stimulus targets. Conversely, most of the mosaic stimuli they had poor performances, excepting Letter X for both subjects and Musical keynote target for Tico.

### Assessment of shape discrimination: Matching-to-sample procedure

Displayed in Table 4 are the subjects' performance results concerning the shape discrimination test, as conducted through the delayed matching-to-sample paradigm. Once more, both

**Table 3. Performance for simple shape discrimination from both subjects.** The statistical significance was based on probability of success = 0.11, number of trials, and number of successes. The values represent the number of successes per number of trials.

| | Raul | | | | Tico | | | |
|---|---|---|---|---|---|---|---|---|
| Stimulus | Solid | p-value | Mosaic | p-value | Solid | p-value | Mosaic | p-value |
| Circle | 73/96 | <0.001* | 1/16 | 0.84 | 79/120 | <0.001* | 0/16 | 1 |
| Square | 90/96 | <0.001* | 4/16 | 0.09 | 85/120 | <0.001* | 0/16 | 1 |
| Letter X | 93/96 | <0.001* | 9/16 | <0.001* | 97/120 | <0.001* | 13/16 | <0.001* |
| Star | 92/96 | <0.001* | 4/16 | 0.09 | 99/120 | <0.001* | 4/16 | 0.09 |
| Musical keynote | 85/96 | <0.001* | 0/16 | 1 | 99/120 | <0.001* | 5/16 | 0.02* |
| Number 8 | 81/96 | <0.001* | 6/16 | 0.055 | 79/120 | <0.001* | 6/16 | 0.055 |

*Significant number of successes.

subjects showcased noteworthy performance levels when discerning shapes depicted as solid figures. Moreover, even when faced with mosaic targets, their capacity for discrimination remained low for most of the targets, but they have again significant discrimination for Letter X and Number 8 targets. Notably, during this context, Tico additionally achieved significant discrimination prowess for the Star shape.

## Discussion

The present investigation reveals that the two capuchin monkeys showcased a limited capacity to consolidate the constituents of mosaic stimuli, displaying a restrained ability to respond guided by local cues. Intriguingly, these very subjects showcased pronounced proficiency in local perceptual processing during the assessment involving hierarchical stimuli, a trend that concurs with findings in the extant literature [5].

Based on the Gestalt principle of similarity, elements sharing attributes such as color, brightness, contrast, and texture tend to coalesce [30]. In the context of this study, the mosaic targets' components possessed luminance similarities. It's imperative to consider the Gestalt principle of proximity, which plays a pivotal role in the process of perceptual grouping [30]. Notably, the haphazard arrangement of elements within the mosaic stimulus plays a crucial role in target appearance, as it incorporates circles with varying luminances, potentially facilitating target perception and localization. Nevertheless, mere proximity between circles of the same luminance failed to ensure successful target shape discrimination. The random spatial distribution of mosaic elements may have contributed to the incomplete emergence of target perception by the two capuchin monkeys. Notably, it seems that the pivotal factor governing

**Table 4. Performance for shape discrimination using matching-to-sample procedure from both subjects.** The statistical significance was based on probability of success = 0.33, number of trials, and number of successes. The values represent the number of successes per number of trials.

| | Raul | | | | Tico | | | |
|---|---|---|---|---|---|---|---|---|
| Stimulus | Solid | p-value | Mosaic | p-value | Solid | p-value | Mosaic | p-value |
| Circle | 44/45 | <0.001* | 15/45 | 0.55 | 44/45 | <0.001* | 13/45 | 0.78 |
| Square | 42/45 | <0.001* | 11/45 | 0.92 | 45/45 | <0.001* | 14/45 | 0.67 |
| Letter X | 39/45 | <0.001* | 30/45 | <0.001* | 32/45 | <0.001* | 22/45 | 0.02* |
| Star | 45/45 | <0.001* | 45/125 | 0.29 | 40/45 | <0.001* | 56/125 | 0.004* |
| Musical keynote | 41/45 | <0.001* | 42/125 | 0.5 | 35/45 | <0.001* | 50/125 | 0.06 |
| Number 8 | 43/45 | <0.001* | 62/125 | <0.001* | 38/45 | <0.001* | 76/125 | <0.001* |

*Significant number of successes.

shape discrimination for both subjects pertains to the shape of certain objects. Intriguingly, although all mosaic targets were detected by the subjects, not all were successfully discriminated based on shape cues, be it in the delayed matching-to-sample task or the simple discrimination procedure. The precise reason behind the subjects' significant discrimination performance for certain shapes while struggling with others remains enigmatic. For instance, both subjects demonstrated poor discrimination skills for rudimentary shapes like circles or squares. Notably, shapes such as stars and musical keynotes were significantly discriminated by Tico (musical keynote in the simple discrimination task, and star in the matching-to-sample procedure), while Raul failed to demonstrate similar aptitude. This disparity suggests that individual factors could be influencing the ability to discriminate these shapes. Intriguingly, both subjects showcased significant shape discrimination prowess when confronted with mosaic targets exhibiting shapes like the Letter X and Number 8 targets. Notably, the shapes of Number 8 and Letter X utilized in the experiments bear a striking resemblance due to their antiparallel diagonal line patterns. The authors have no suggestion about why specifically both targets elicited significant shape discrimination for both animals.

To date, the authors are unaware of any studies in non-human primates equivalent to the present study that have examined shape discrimination using mosaics. There is at least one article with humans in which the participant was asked to perform a task in which the shape of the target seen in the mosaic was compared with targets present in solid stimuli [31] although shape discrimination was not the focus of the study, which sought to estimate chromatic discrimination thresholds. The closest studies with nonhuman primates that we can relate to our findings are shape discrimination experiments using Kanizsa illusory shapes [32]. In these experiments, as in the experiment in our study, it was hypothesized that the emergence of illusory shapes would be due to global processing of the image, and that the animals' good performance in these tasks would be a strong indicator that global processing of the visual scene was taking place. It should be noted that in these experiments, as in ours, there was variability in the animals' performance in recognizing the illusory shapes.

It's important to note that within the global and local processing hierarchical experiments, where the stimuli featured uniformly spaced elements composing the target, global perception did not manifest. The discerned prevalence of local control's advantage within hierarchical stimuli, as revealed in this study, aligns cohesively with analogous findings in capuchin monkeys from other studies [1, 6, 14, 16, 18, 19], as well as in cotton top tamarins [4].

The present study diverges from earlier research examining the impact of local and global cues on image perceptual organization, owing to its utilization of mosaic stimuli. Unlike the hierarchically structured stimuli traditionally employed in such inquiries, mosaic stimuli introduce a significantly heightened level of complexity and, notably, have never been previously applied for this specific purpose. While mosaic stimuli have extensively found utility in appraising the color vision phenotype of non-human primates, their novel application in tasks necessitating shape discrimination is unprecedented. Mosaic stimuli present a multifaceted array of variables that can be manipulated within these tests, including the elemental dimensions constituting the mosaic, the spacing separating these elements, and diverse measures of color and luminance contrast. Therefore, a more comprehensive future investigation, exploring the impact of different stimulus parameters, could provide valuable insights into the underlying mechanisms of shape perception in capuchin monkeys. In the way these are put, it turns out to be mostly confounding variables in this study.

Our findings suggest that the monkeys' responses are influenced by local contrast and, to a certain extent, by global contrast within the mosaic stimulus. However, it is plausible that additional manipulations employing mosaic stimuli would be necessary to definitively establish the potential for monkeys to exhibit more refined global perceptual grouping. An illustrative study

conducted by Han et al. [33] explored neural grouping mechanisms rooted in proximity and similarity, leveraging brain event-related potentials (ERPs) as recorded from human participants. The results of their investigation unveiled that participants exhibited quicker and more accurate perceptions of grouping based on proximity compared to similarity. The ERP data revealed a brief latency positivity in the occipital cortex corresponding to proximity and a prolonged latency occipitotemporal negativity indicative of similarity. The authors posit the prevalence of proximity over similarity in the local element grouping for perceptual shape perception in humans. However, it remains essential to probe this issue within capuchin monkeys, as the existence of analogous mechanisms for perceptual grouping between humans and capuchins is far from guaranteed.

The congruence in results observed across the two subjects underscores the consistency between methodologies and outcomes. However, the limitation of a small subject pool in this study necessitates prudence in drawing comprehensive conclusions. To enhance the robustness of our findings, future investigations could encompass a larger sample size of monkeys, potentially including females and infants. The need for further exploration into the realm of visual processing within capuchin monkeys remains evident to fathom the intricacies of their visual cognition. The present study's results offer preliminary indications that capuchin monkeys potentially discern shapes arising from the grouping of elements by luminance similarity within mosaics. This insight stands poised to guide forthcoming studies probing perceptual grouping employing mosaic stimuli. Importantly, such investigations must account for the distinct sensitivity to proximity and similarity between elements unique to each species.

## Supporting information

**S1 Data.**
(XLSX)

## Author Contributions

**Conceptualization:** Fernanda Mendes, Ana Leda de Faria Brino, Paulo Roney Kilpp Goulart.

**Data curation:** Fernanda Mendes, Ana Leda de Faria Brino, Givago Silva Souza.

**Formal analysis:** Fernanda Mendes, Ana Leda de Faria Brino, Givago Silva Souza.

**Funding acquisition:** Fernanda Mendes, Ana Leda de Faria Brino, Olavo de Faria Galvão, Dora Selma Fix Ventura, Givago Silva Souza.

**Investigation:** Fernanda Mendes, Ana Leda de Faria Brino.

**Methodology:** Fernanda Mendes, Ana Leda de Faria Brino, Paulo Roney Kilpp Goulart, Olavo de Faria Galvão, Givago Silva Souza.

**Project administration:** Fernanda Mendes, Ana Leda de Faria Brino, Givago Silva Souza.

**Resources:** Ana Leda de Faria Brino.

**Software:** Paulo Roney Kilpp Goulart, Felipe André da Costa Brito, Givago Silva Souza.

**Supervision:** Ana Leda de Faria Brino, Givago Silva Souza.

**Validation:** Fernanda Mendes, Ana Leda de Faria Brino, Paulo Roney Kilpp Goulart, Olavo de Faria Galvão, Dora Selma Fix Ventura, Letícia Miquilini, Givago Silva Souza.

**Visualization:** Fernanda Mendes.

**Writing – original draft:** Fernanda Mendes, Ana Leda de Faria Brino.

**Writing – review & editing:** Fernanda Mendes, Ana Leda de Faria Brino, Paulo Roney Kilpp Goulart, Olavo de Faria Galvão, Dora Selma Fix Ventura, Letícia Miquilini, Felipe André da Costa Brito, Givago Silva Souza.

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
