## [Decision Letter · Decision Letter 0]

15 Feb 2024

PONE-D-24-01574Shape Discrimination in Mosaic Designs in Capuchin-Monkeys (*Sapajus* spp.): Indication of a preference for global processing?PLOS ONE

Dear Dr. Mendes,

Thank you for submitting your manuscript to PLOS ONE. After careful consideration, we feel that it has merit but does not fully meet PLOS ONE’s publication criteria as it currently stands. Therefore, we invite you to submit a revised version of the manuscript that addresses the points raised during the review process.

We look forward to receiving your revised manuscript.

Kind regards,

Carlos Tomaz, Ph.D.

Academic Editor

PLOS ONE

Journal Requirements:

"DFV, GSS, OFG

#431748/2016-0 

Conselho Nacional de Desenvolvimento Científico e Tecnológico (CNPq) 

https://www.gov.br/cnpq/pt-br

Funding through

FM, FACB, LM

Finance Code 001

Coordenação de Aperfeiçoamento de Pessoal de Nível Superior (CAPES) 

" ext-link-type="uri" xlink:type="simple">https://www.gov.br/capes/pt-br"

Please state what role the funders took in the study.  If the funders had no role, please state: ""The funders had no role in study design, data collection and analysis, decision to publish, or preparation of the manuscript."" If this statement is not correct you must amend it as needed. 

3. We note that your Data Availability Statement is currently as follows: "All relevant data are within the manuscript and its Supporting Information files."

Reviewers' comments:

Reviewer's Responses to Questions

**Comments to the Author**

1. Is the manuscript technically sound, and do the data support the conclusions?

Reviewer #1: No

Reviewer #2: Yes

2. Has the statistical analysis been performed appropriately and rigorously? 

Reviewer #1: Yes

Reviewer #2: I Don't Know

3. Have the authors made all data underlying the findings in their manuscript fully available?

Reviewer #1: Yes

Reviewer #2: Yes

4. Is the manuscript presented in an intelligible fashion and written in standard English?

Reviewer #1: Yes

Reviewer #2: Yes

5. Review Comments to the Author

Reviewer #1: The manuscript evaluated shape discrimination capabilities of capuchin monkeys to gauge the dominance of local and global processing. Its main difference from previous studies seems to be on the type of stimuli used, such as mosaic stimuli. Results suggest that the monkeys primarily respond to local contrast and partly to global contrast in mosaic stimuli, which seems to differ from other studies using other classical paradigms.

The authors acknowledge the limitations of the study, especially that the results are only suggestive and inconclusive. But I believe rather than being an issue with the number of subjects, there are issues with the stimuli used. I was not able to understand what mosaic stimuli consisted of (perhaps the figure is not big enough also), and it is not clear if it’s a new paradigm or based on previous studies, since the references mentioned in comparison (19, 11, 13, and 14) don’t call their stimuli as mosaic (and also don’t seem to be the same type as the current study). Given this is an important feature of this study, more clarification is needed into what this stimulus is, as well as its importance in visual processing and in this study itself.

Another small issue is the different measures given for the stimuli, with some features measured in arc-seconds (degrees?), some in pixels, and others in cm. Some consistency could help with replication.

I commend the author's thorough description of some technicalities. Despite that, a few parts may not be relevant, such as the processor of the computer used (line 123), and mentioning the software “Microsoft paint" twice through the text. But I would like to make clear this does not make the manuscript any poorer and I’m not basing my evaluation on it, just as a suggestion.

One technical detail that does need better explanation is the order of the stimuli shown. Is it randomized, always the same order, and is it the same for both subjects?

In Experimental Procedures, line 133, there is again a reminder on the manuscript’s focus on the mosaic designs, but it is not clear through the paper the reasoning behind the choice of all these stimuli paired with the mosaic design.

In the explanation of the experiment in lines 185-204 it’s not clear if every session follows that order or if changes between sessions/subjects.

In line 214 should it be the third experiment? The inconsistent use of each of the 3 experiment names through the manuscript (for example between line 261 and 268) makes it sometimes hard to follow.

Are the outlier performances of each subject in specific shapes (lines 285-286 and 296-297) of any significance to the results? If so it would be better mentioned in the discussion. if not, either just grouped as outlier performances without known reason by the authors, or removed since it can be seen on the tables already.

As a general limitation, the authors acknowledged that mosaic stimuli involve various factors that could influence the monkeys' responses, such as the elemental dimensions, spacing, and color contrast. However, they did not systematically manipulate many of these variables to determine their specific effects on shape discrimination. A more comprehensive investigation, exploring the impact of different stimulus parameters, would have provided valuable insights into the underlying mechanisms of shape perception in capuchin monkeys. In the way these are put, it turns out to be mostly confounding variables in this study.

There is also a lack of comparisons between their findings with other research examining shape discrimination in mosaic stimuli in either human or non-human populations. This omission makes it challenging to contextualize the results and determine whether the observed patterns are unique to capuchin monkeys or are consistent with broader trends in visual perception. Including data from other studies would have provided a more comprehensive perspective and strengthened the conclusions.

As a whole, the manuscript needs to be more forthcoming as an exploratory research and better interconnect with other studies, or improve their experimental paradigm to give more conclusive objective results.

Reviewer #2: Thank you for the opportunity to review this manuscript, which details 4 screen-based studies of visual cognition in two captive capuchin monkeys. The manuscript provides key information about methods (including animal housing, ethical review, the experimental chamber and the touchscreen); and the figures and tables are useful (it is nice to see the experimental chamber, and examples of the stimuli help the reader understand the conditions and procedures). I think these studies have the potential to make a valuable contribution to the literature, once some questions are addressed.

First, in general throughout the manuscript it is not clear what is meant by ‘mosaic stimuli’. (At Line 71, it is stated that this type of stimulus is common in chromatic discrimination tasks.) From the figures provided, it appears that the mosaic stimuli are simply on a darker gray background than the linear images. It would be very helpful to describe these stimuli and give a clear exemplar in a figure early on in the manuscript (especially as it is claimed in the Discussion that this is a novel use of this type of stimulus). It may also be helpful to use a different name for them, like ‘dot stimuli’ or ‘dot-mosaic stimuli’, or similar (as previous examples of the term ‘mosaic stimuli’ in the literature do not always describe the same type of stimulus).

Second, as it stands I do not believe there is enough information in the Methods to replicate the procedure. Therefore, in some places more experimental detail is needed in order for the reader to understand the experiments. For example, at Lines 158-162 (matching to sample of hierarchical stimuli): How was it indicated to the monkeys which was the global and which the local condition? Were they visually identical other than the stimulus that was rewarded, or did they have different backgrounds (or other feature) that would let the monkey know which condition they were performing? Which stimuli were included in each condition (e.g., were there only 2 possible stimuli for each condition, and were these alternated as targets in some sort of pattern?)? How long was the delay between initial selection of the target and the availability of the responses? Line 158 states that this was ‘training’: were there separate test sessions also? It is stated (Lines 161-162) that there were 5 sessions with 12 trials each (60 trials) for each condition; in Table 1, each monkey appears to have received 120 trials in each condition. Can you please clarify? At Lines 188-189 (target discrimination): Were the different shapes in the target discrimination trials randomized throughout a session, or were they presented in groups?

Third, a suggestion for readability would be to combine the Methods Results for each experiment, as otherwise it can be hard to keep track of what’s happening.

Individual points

Title: Based on the results of the studies taken as a whole, the title’s second half (“Indication of a preference for local processing?”) doesn’t effectively summarize the findings.

Throughout manuscript: I believe it is now customary to refer to ‘New World monkeys’ as ‘platyrrhine monkeys’: would you be willing to adjust?

A specific example of a hierarchical stimulus (even just described in the text) at Lines 54-55 would be helpful to the reader.

Lines 60-61: can you clarify what you mean by ‘for inconsistent stimuli, the global level resulted in slower responses to the local level, but not vice versa’?

Line 64 (also Lines 337-338): the referenced paper’s title states that it is about ‘cotton-top tamarins’ (Saguinus oedipus) rather than ‘cotton-headed marmosets’. Please clarify.

Lines 88-89: Can you clarify whether monkeys’ experience with discrimination and MTS were with using a touchscreen?

Line 120: Which type of touch technology does the touchscreen use? (capacitive, resistive, infrared, other?)

Can you provide a statement about whether animals were food-deprived or kept at a certain percentage of free-feeding weight (or not) for the purposes of the study?

Lines 148-155 (Figure 2 legend?): S+ and S- are not mentioned in the figure.

For all tables: please indicate what the asterisks mean. If they indicate ‘above-chance performance’, can you please also state the chance level for each task in the table or legend somewhere?

Table 2 and 3: is there a reason that Raul had more solid target discrimination trials than Tico? And that Tico had more shape discrimination trials than Raul?

Line 232: Here you mention ‘gray fields’, but the shape discrimination experiment to my understanding did not contain gray fields: please clarify.

Lines 248-249: It is stated that ‘The performance criterion for terminating a session was either achieving a 90% accuracy rate or the completion of 5 sessions.’ Can you clarify? (Would a monkey have performed more than one 54-trial session in a single day?)

Line 250-255: Can you please clarify which the statistical tests you conducted, and how they were implemented (i.e., which statistical programme)?

In the discussion (Lines 341-345) it is stated that this is a novel application of mosaic stimuli. This information could usefully be introduced in the Introduction section.

6. PLOS authors have the option to publish the peer review history of their article (what does this mean?). If published, this will include your full peer review and any attached files.

Reviewer #1: No

Reviewer #2: No

---

## [Author Response · Author response to Decision Letter 0]

8 Apr 2024

Reviewer #1:

The manuscript evaluated shape discrimination capabilities of capuchin monkeys to gauge the dominance of local and global processing. Its main difference from previous studies seems to be on the type of stimuli used, such as mosaic stimuli. Results suggest that the monkeys primarily respond to local contrast and partly to global contrast in mosaic stimuli, which seems to differ from other studies using other classical paradigms.

The authors acknowledge the limitations of the study, especially that the results are only suggestive and inconclusive. But I believe rather than being an issue with the number of subjects, there are issues with the stimuli used. I was not able to understand what mosaic stimuli consisted of (perhaps the figure is not big enough also), and it is not clear if it’s a new paradigm or based on previous studies, since the references mentioned in comparison (19, 11, 13, and 14) don’t call their stimuli as mosaic (and also don’t seem to be the same type as the current study). Given this is an important feature of this study, more clarification is needed into what this stimulus is, as well as its importance in visual processing and in this study itself.

A. Thanks for the comment. We wrote a more detailed explanation of our choice of using mosaic arrangement to investigate the preference of global or local processing. Additionally, we added a Figure with an example of mosaic arrangement and more detailed paragraph describing why mosaic arrangements could be important to investigate the global and local processing.

Another small issue is the different measures given for the stimuli, with some features measured in arc-seconds (degrees?), some in pixels, and others in cm. Some consistency could help with replication.

A. Done.

I commend the author's thorough description of some technicalities. Despite that, a few parts may not be relevant, such as the processor of the computer used (line 123)

A. Right. We excluded the information.

, and mentioning the software “Microsoft paint" twice through the text. But I would like to make clear this does not make the manuscript any poorer and I’m not basing my evaluation on it, just as a suggestion.

A. We wrote again both parts. Thanks.

One technical detail that does need better explanation is the order of the stimuli shown. Is it randomized, always the same order, and is it the same for both subjects?

A. Thanks for the comment. We included that for each experiment the order of stimulus presentation was randomly chosen by the software.

In Experimental Procedures, line 133, there is again a reminder on the manuscript’s focus on the mosaic designs, but it is not clear through the paper the reasoning behind the choice of all these stimuli paired with the mosaic design.

A. We added to the introduction the rationale for the choice of the mosaic stimuli, and we hope to make clearer our purpose.

In the explanation of the experiment in lines 185-204 it’s not clear if every session follows that order or if changes between sessions/subjects.

A. We rewrote the sentence with “The same order of sessions was delivered for both animals. Inside each session the order of stimulus presentation was randomly chosen by the software.”

In line 214 should it be the third experiment? The inconsistent use of each of the 3 experiment names through the manuscript (for example between line 261 and 268) makes it sometimes hard to follow.

A. Thanks again. We searched for to name correctly the experiments in the revised version.

Are the outlier performances of each subject in specific shapes (lines 285-286 and 296-297) of any significance to the results? If so it would be better mentioned in the discussion. if not, either just grouped as outlier performances without known reason by the authors, or removed since it can be seen on the tables already.

A. Thanks for the suggestion. We stated in the Discussion that both outlier performances were without know by us.

As a general limitation, the authors acknowledged that mosaic stimuli involve various factors that could influence the monkeys' responses, such as the elemental dimensions, spacing, and color contrast. However, they did not systematically manipulate many of these variables to determine their specific effects on shape discrimination. A more comprehensive investigation, exploring the impact of different stimulus parameters, would have provided valuable insights into the underlying mechanisms of shape perception in capuchin monkeys. In the way these are put, it turns out to be mostly confounding variables in this study.

A. Thanks for the comment. We added this observation in the Discussion section.

There is also a lack of comparisons between their findings with other research examining shape discrimination in mosaic stimuli in either human or non-human populations. This omission makes it challenging to contextualize the results and determine whether the observed patterns are unique to capuchin monkeys or are consistent with broader trends in visual perception. Including data from other studies would have provided a more comprehensive perspective and strengthened the conclusions.

A. Done.

As a whole, the manuscript needs to be more forthcoming as an exploratory research and better interconnect with other studies, or improve their experimental paradigm to give more conclusive objective results.

Reviewer #2:

Thank you for the opportunity to review this manuscript, which details 4 screen-based studies of visual cognition in two captive capuchin monkeys. The manuscript provides key information about methods (including animal housing, ethical review, the experimental chamber and the touchscreen); and the figures and tables are useful (it is nice to see the experimental chamber, and examples of the stimuli help the reader understand the conditions and procedures). I think these studies have the potential to make a valuable contribution to the literature, once some questions are addressed.

First, in general throughout the manuscript it is not clear what is meant by ‘mosaic stimuli’. (At Line 71, it is stated that this type of stimulus is common in chromatic discrimination tasks.) From the figures provided, it appears that the mosaic stimuli are simply on a darker gray background than the linear images. It would be very helpful to describe these stimuli and give a clear exemplar in a figure early on in the manuscript (especially as it is claimed in the Discussion that this is a novel use of this type of stimulus). It may also be helpful to use a different name for them, like ‘dot stimuli’ or ‘dot-mosaic stimuli’, or similar (as previous examples of the term ‘mosaic stimuli’ in the literature do not always describe the same type of stimulus).

A. We rewrote the Introduction searching for clarifying the rationale of using mosaic stimulus.

Second, as it stands I do not believe there is enough information in the Methods to replicate the procedure. Therefore, in some places more experimental detail is needed in order for the reader to understand the experiments. For example, at Lines 158-162 (matching to sample of hierarchical stimuli): How was it indicated to the monkeys which was the global and which the local condition? Were they visually identical other than the stimulus that was rewarded, or did they have different backgrounds (or other feature) that would let the monkey know which condition they were performing?

A. The monkeys were instructed on which global and local clues to follow using a reward system that favored each condition.

Which stimuli were included in each condition (e.g., were there only 2 possible stimuli for each condition, and were these alternated as targets in some sort of pattern?)?

A. There were only two possible stimuli.

How long was the delay between initial selection of the target and the availability of the responses? 

A. 1 second. We added to the text.

Line 158 states that this was ‘training’: were there separate test sessions also? It is stated (Lines 161-162) that there were 5 sessions with 12 trials each (60 trials) for each condition; in Table 1, each monkey appears to have received 120 trials in each condition. Can you please clarify?

A. There was no test. The animals were exposed to five sessions with global configuration trials and another five sessions with local configuration trials. The trials were repeated and there were differential consequences for success and error. Five 48-trial sessions had global configurations, in a total of 240 trials, 120 of global consistent and 120 of local consistent; the same occurred for the five 48-trial sessions of local configuration, 240 total trials, 120 local consistent and 120 global inconsistent.

At Lines 188-189 (target discrimination): Were the different shapes in the target discrimination trials randomized throughout a session, or were they presented in groups?

A. It was randomized.

Third, a suggestion for readability would be to combine the Methods Results for each experiment, as otherwise it can be hard to keep track of what’s happening.

A. Done.

Individual points

Title: Based on the results of the studies taken as a whole, the title’s second half (“Indication of a preference for local processing?”) doesn’t effectively summarize the findings.

A. We changed the title for Investigation of preference for local and global processing of Capuchin-Monkeys (Sapajus spp.) in shape discrimination in mosaic arrangements

Throughout manuscript: I believe it is now customary to refer to ‘New World monkeys’ as ‘platyrrhine monkeys’: would you be willing to adjust?

A. Done.

A specific example of a hierarchical stimulus (even just described in the text) at Lines 54-55 would be helpful to the reader.

A. Done. We included a figure with examples of hierarchical stimuli in the Introduction section.

Lines 60-61: can you clarify what you mean by ‘for inconsistent stimuli, the global level resulted in slower responses to the local level, but not vice versa’?

A. We excluded the sentence.

Line 64 (also Lines 337-338): the referenced paper’s title states that it is about ‘cotton-top tamarins’ (Saguinus oedipus) rather than ‘cotton-headed marmosets’. 

Please clarify.

A. Thank you very much for the comment. We have changed the name.

Lines 88-89: Can you clarify whether monkeys’ experience with discrimination and MTS were with using a touchscreen?

A. Both subjects had extensive experience (at least 10 years) with discrimination and MTS were with using a touchscreen (references).

Line 120: Which type of touch technology does the touchscreen use? (capacitive, resistive, infrared, other?)

A. It was used a surface acoustic wave touchscreen-type device.

Can you provide a statement about whether animals were food-deprived or kept at a certain percentage of free-feeding weight (or not) for the purposes of the study?

A. Before the experiments the subject were kept at a certain percentage of free-feeding weight.

Lines 148-155 (Figure 2 legend?): S+ and S- are not mentioned in the figure.

A. Thanks. Done.

For all tables: please indicate what the asterisks mean. If they indicate ‘above-chance performance’, can you please also state the chance level for each task in the table or legend somewhere?

A. Done.

Table 2 and 3: is there a reason that Raul had more solid target discrimination trials than Tico? And that Tico had more shape discrimination trials than Raul?

A. Thanks for the questions. Raul had reached performance criterion before Tico. The total number of trials with solid stimuli was lower for Raul (96 trials distributed in 12 sessions) than for Tico (120 trials distributed in 15 sessions). We wrote again the sentence and we informed that was tried several sessions up to reach 90% correct in one session and when the performance was lower than this criterion in 10 consecutive sessions, the test was completed. Each subject performed one session (48 trials) per day.

Line 232: Here you mention ‘gray fields’, but the shape discrimination experiment to my understanding did not contain gray fields: please clarify.

A. Oh really, we replaced the gray fields by S-.

Lines 248-249: It is stated that ‘The performance criterion for terminating a session was either achieving a 90% accuracy rate or the completion of 5 sessions.’ Can you clarify? (Would a monkey have performed more than one 54-trial session in a single day?)

A. Thanks for the question. We wrote again the sentence and we informed that was tried several sessions up to reach 90% correct in one session and when the performance was lower than this criterion in 10 consecutive sessions, the test was completed. The subject was exposed to one session per day and the session was composed by 48 trials. 

Line 250-255: Can you please clarify which the statistical tests you conducted, and how they were implemented (i.e., which statistical programme)?

A. Thanks. We added that we wrote a routine in Python language to calculate the statistical significance of the subject’s performance in each experiment by calculating the binomial probability

In the discussion (Lines 341-345) it is stated that this is a novel application of mosaic stimuli. This information could usefully be introduced in the Introduction section.

A. Done

---

## [Decision Letter · Decision Letter 1]

29 Apr 2024

Investigation of preference for local and global processing of Capuchin-Monkeys (*Sapajus* spp.) in shape discrimination of mosaic arrangements

PONE-D-24-01574R1

Dear Dr. Mendes,

We’re pleased to inform you that your manuscript has been judged scientifically suitable for publication and will be formally accepted for publication once it meets all outstanding technical requirements.

Kind regards,

Carlos Tomaz, Ph.D.

Academic Editor

PLOS ONE

Additional Editor Comments (optional):

The authors have addressed all the comments and suggestions from the referees. Therefore, I understand that the MS can be now be accept for publication.

Reviewers' comments:

Reviewer's Responses to Questions

**Comments to the Author**

1. If the authors have adequately addressed your comments raised in a previous round of review and you feel that this manuscript is now acceptable for publication, you may indicate that here to bypass the “Comments to the Author” section, enter your conflict of interest statement in the “Confidential to Editor” section, and submit your "Accept" recommendation.

Reviewer #1: All comments have been addressed

Reviewer #2: (No Response)

2. Is the manuscript technically sound, and do the data support the conclusions?

Reviewer #1: Yes

Reviewer #2: Yes

3. Has the statistical analysis been performed appropriately and rigorously? 

Reviewer #1: I Don't Know

Reviewer #2: Yes

4. Have the authors made all data underlying the findings in their manuscript fully available?

Reviewer #1: No

Reviewer #2: Yes

5. Is the manuscript presented in an intelligible fashion and written in standard English?

Reviewer #1: No

Reviewer #2: Yes

6. Review Comments to the Author

Reviewer #1: Thank you for directly addressing all the comments from the previous round of review.

The methods appear correct and the new information added enriches the manuscript and imparts importance to the study by making easier to compare to previous (and future) studies.

I'm specially satisfied with the new figures and clear explanation of the experiment and it's importance.

Despite that, I believe English editing is necessary in the manuscript in the current form, since Plos One does not perform language editing pre-publication. Besides a few grammatical mistakes, some parts make hard to understand what the authors intended to say.

Although this comment is mostly directed to the newly added parts, I must say it concerns me that some sentences and paragraphs appear disjointed from each other. I don't believe this issue is serious enough to compromise the scientific interpretation of the paper, but it is noticeable and could be easily fixed with an English revision.

Reviewer #2: Thank you for the opportunity to read and review this revised manuscript about capuchin monkeys’ visual cognition. The authors have addressed the majority of my questions in their responses and in the revised manuscript, which is improved in readability and clarity. Thank you also for providing the data file in Supporting Information. Figure 1 and the revised Figure 3 are nice additions/edits.

A few minor issues are outstanding.

By “significant” performance levels (throughout the manuscript), do you mean that their performance was better or worse than chance?

It is stated that the monkeys were kept at a certain percentage of free-feeding weight: which percentage? (i.e., just to confirm, they were food deprived to increase motivation?)

Were the statistical tests binomial tests?

Please check the sentences at lines 256-257 and 294-297; they could be clearer.

While the two experiments described together for Experiment 3 (shape discrimination and MTS) have a great deal in common, I wonder if they also could be described separately for clarity.

Check Figure 5 – the mosaic images are fairly dark in the PDF, but I wonder if this is due to the PDF-builder rather than an issue with the actual image file. *FOR EDITOR TO CHECK?

7. PLOS authors have the option to publish the peer review history of their article (what does this mean?). If published, this will include your full peer review and any attached files.

Reviewer #1: No

Reviewer #2: No

---

## [Editor Report · Acceptance letter]

8 May 2024

PONE-D-24-01574R1 

PLOS ONE

Dear Dr. Mendes, 

I'm pleased to inform you that your manuscript has been deemed suitable for publication in PLOS ONE. Congratulations! Your manuscript is now being handed over to our production team.

Kind regards, 

on behalf of

Dr. Carlos Tomaz 

Academic Editor

PLOS ONE